# Perioperative patient safety management activities: A modified theory of planned behavior

**Nam Yi Kim**[1], **Sun Young Jeong**[2]*

1 Department of Nursing, Daejeon Institute of Science and Technology, Daejeon, Republic of Korea,
2 College of Nursing, Konyang University, Daejeon, Republic of Korea

* jsy7304@konyang.ac.kr

## Abstract

Patient safety is an important healthcare issue worldwide, and patient accidents in the operating room can lead to serious problems. Accordingly, we investigated the explanatory ability of a modified theory of planned behavior to improve patient safety activities in the operating room. Questionnaires were distributed to perioperative nurses working in 12 large hospitals in Korea. The modified theory of planned behavior data from a total of 330 nurses were analyzed. The conceptual model was based on the theory of planned behavior data, with two additional organizational factors—job factors and safety management system. Individual factors included attitude, subjective norms, perceived behavioral control, behavioral intention, and patient safety management activities. Results indicated that job factors were negatively associated with perceived behavioral control. The patient safety management system was positively associated with attitude, subjective norm, and perceived behavioral control. Attitude, subjective norm, and perceived behavioral control were positively associated with behavioral intention. Behavioral intention was positively associated with patient safety management activities. The modified theory of planned behavior effectively explained patient safety management activities in the operating room. Both organizations and individuals are required to improve patient safety management activities.

## Introduction

With multidisciplinary professionals, diverse and complex medical equipment, vulnerable patients, time pressures, and extremely high tension, the operating room (OR) environment is susceptible to errors [1]. Some major safety problems in the OR include addressing incorrect surgical site/patient/procedure, retained surgical items, medication errors, bedsores, hypothermia, burns, inadequate emergency responses, and improperly reprocessing surgical devices [2]. Patient safety accidents related to surgery require particular precautions, as they can induce serious and irreversible injuries [3]. Hence, the Joint Commission on Accreditation of Healthcare Organization (JCAHO) stressed the importance of teamwork, continuous quality control, smooth communication, and information sharing between medical professionals to ensure surgical patients' safety [4]. Furthermore, the Association of periOperative Registered Nurses (AORN) recommended quickly streamlining and standardizing work to detect and correct errors that occur during surgery [5].

researchers who meet the criteria for access to confidential data. Data requests can be addressed to the Konyang University Hospital Institutional Review Board (82-42-600-9057, leesh@kyuh.ac. kr). Researchers may reference our data set with the following title: 'Perioperative Patient Safety Management Activities'

**Funding:** The authors received no specific funding for this work.

**Competing interests:** The authors have declared that no competing interests exist.

Despite such efforts, accidents continue to occur with surgical patients. According to a systematic literature review of adverse events in hospitals, surgery-related accidents accounted for 39.6%, the highest proportion of all such events [6]. In Korea, surgery-related cases accounted for the highest proportion (35.1%) of all medical dispute claims filed between 2012 and 2016 and are gradually increasing [7]. Therefore, surgical patients' safety is of utmost importance.

## Literature review

The theory of planned behavior (TPB) describes individual-level predictors of actions [8]. This theory states that individuals' conduct consists of their attitudes toward behavior, subjective norms, perceived behavioral control, and behavioral intention [9]. The TPB is widely used not only in social sciences but also in various healthcare fields, as it effectively predicts individuals' behavior despite involving only a few simple constructs. However, no studies have applied TPB to patient safety management activities. Accordingly, we applied Ajzen's TPB to establish a model for patient safety management activities in the OR.

Human errors can be viewed at the individual or system level. System-level human errors are lapses in safety behaviors attributed to conditions of the work environment, which cannot be altered by an individual [10]. We need to understand how systems, which include organizational culture and policies, interact with individuals. Thus, patient safety management activities in the OR should be examined considering both individual and organizational factors, given that social behaviors result from their interactions.

The most frequently examined organizational factors related to patient safety are job and systemic factors such as the safety management system [11,12]. Job factors include excessive work demands and job complexity, which increase the physical and cognitive burdens on healthcare professionals, thereby decreasing their ability to engage in safety management activities [13,14]. The safety management system includes safety training, participation in safety policy, management supervision, communication, and feedback. Therefore, when the level of safety management system is insufficient, accurate information on safety is not delivered, and education and management are neglected, thus resulting in lower awareness and performance of patient safety management activities [15–17].

Therefore, we developed a hypothetical model that encompasses individual and system dimensions of safety management activities in the OR by adding organizational factors (i.e., job factors and safety management system) to Ajzen's TPB (Fig 1). Structural modeling studies investigating factors related to operating room safety management activities will be useful in developing effective strategies to enhance patient safety management activities of OR nurses.

### Study aim

The objectives of this study were to develop a structural model for patient safety management activities, identify the factors influencing organizational and individual dimensions that promote patient safety management activities, and suggest effective intervention plans.

## Materials and methods

### Design

A cross-sectional research design was used. A hypothetical model was developed based on Ajzen's TPB. Job factors and the safety management system were used as organizational factors, and attitude, subjective norms, perceived behavioral control, behavioral intention, and patient safety management activities were used as individual factors (Fig 1).

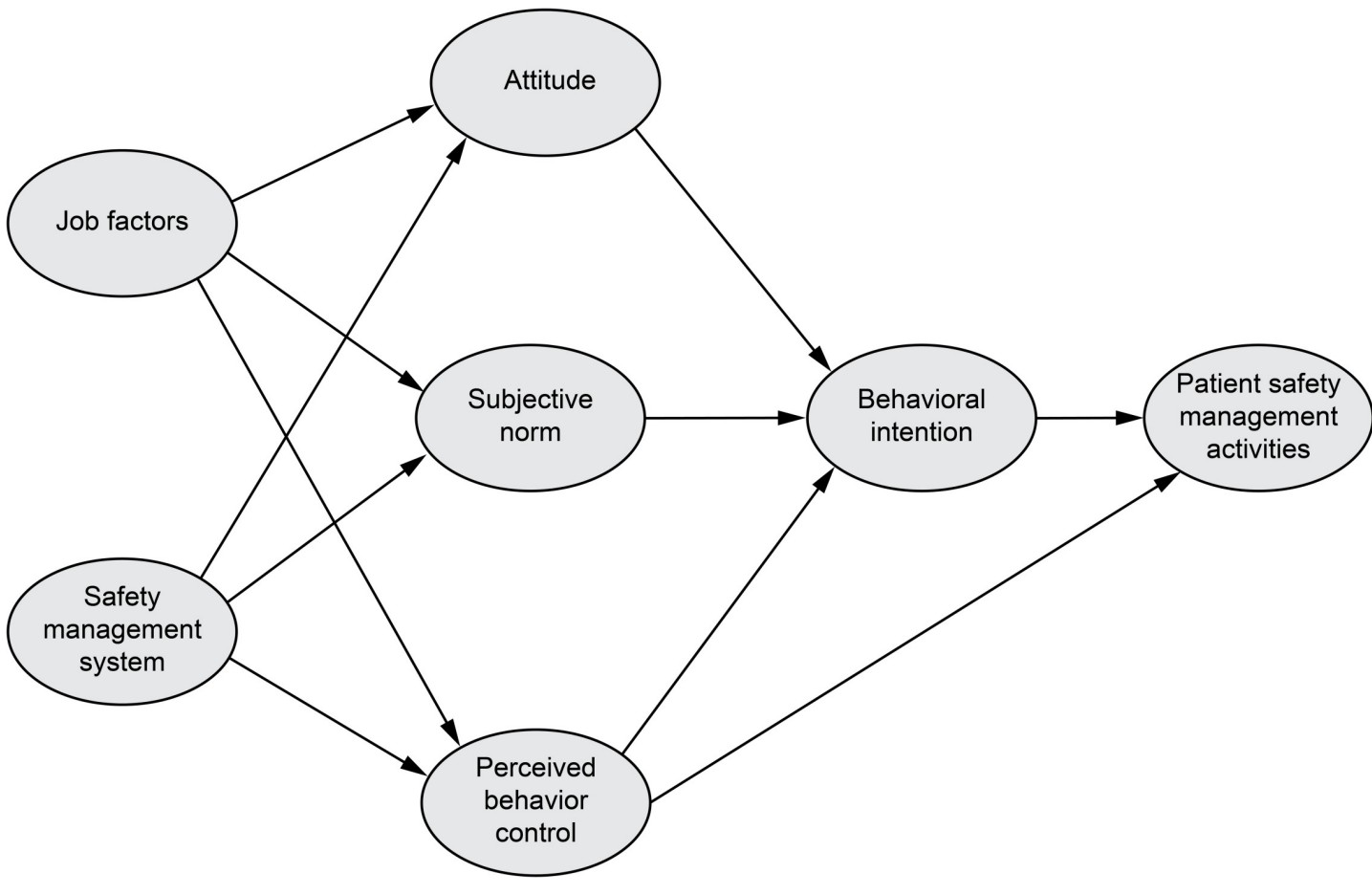

**Fig 1. Conceptual model.**

## Participants and data collection

Data was collected from August 1 to October 31, 2017, using self-report questionnaires. The recommended sample size in the structural equation model was 10–20 per observation variable [18]. The expected number of observation variables was 32; thus, 320 to 640 participants were required. The questionnaires were distributed by convenience sampling to 360 perioperative nurses in 12 general hospitals in the Republic of Korea, and 347 questionnaires were returned (response rate = 96.4%). Questionnaires that were missing >10% of responses were excluded (17 questionnaires). The remaining questionnaires were processed with mean substitution [19]. Consequently, 330 questionnaires were included in the final analyses.

## Instruments

**Job factors.** The Job Content Questionnaire developed by Karasek et al. [20] is a commonly used instrument to assess organizational job factors, and its validity and reliability were verified in a previous study by Song [21], which examined job factors for Korean nurses. Therefore, in this study, the instrument modified by Song [21] for perioperative nurses was used. As for job demand, each of the 10 items in the questionnaire was measured on a five-point Likert scale. Cronbach's alpha was .87 in both Song's [21] study and the current study.

**Safety management system.** Safety management system was evaluated using an instrument developed by Vredenburgh [22] and translated and adapted for use in the OR by Song and Jang [21]. This included management supervision, communication and feedback, and participation system. The 9 items in the scale were measured using a five-point Likert scale. Cronbach's alpha was .73 in Song and Jang's [21] study and .82 in the current study.

**Attitude, subjective norms, and perceived behavioral control.** A 12-item scale was developed based on Ajzen's [23] study and a scale developed by Moon and Song [24] for hospital nurses. Attitude was measured with three questions regarding positive or negative feelings about certain behaviors. The subjective norm is the perceived social pressure imposed on conduct, and was measured using five questions. Perceived behavioral control is an individual's confidence or controllability of a behavior, and was measured with four questions. Each item was rated on a seven-point Likert scale. Cronbach's alpha for attitude was .69 in Moon and Song's [24] study and .77 in the current study. Cronbach's alpha for subjective norms was .76 in Moon and Song's [24] study and .91 in the current study. Similarly, for perceived behavioral control, Cronbach's alpha was .81 in Moon and Song's [24] study and .88 in the current study.

**Behavioral intention.** A 4-item scale was developed based on Ajzen's [23] study and a scale developed by Moon and Song [24] for hospital nurses. Each item was rated on a 7-point Likert scale for willingness, planning, and thinking. Cronbach's alpha for behavioral intention was .80 in Moon and Song's [24] study and .90 in the current study.

**Patient safety management activities.** Safety management activities were measured using an instrument developed by Kim and Jeong [25] based on six international patient safety goals [26]. The scale consisted of items pertaining to infection management, specimen management, patient identification, medical equipment and product management, surgical counting, and injury prevention. The 36 items were measured on a five-point Likert scale. Cronbach's alpha was .95 in Kim and Jeong's study [25] and .94 in the current study.

## Data analyses

Collected data were analyzed using SPSS 25.0 (SPSS; IBM, Armonk, NY, USA) and AMOS 21.0 (SPSS Amos; IBM, Chicago, IL, USA). Participants' general characteristics were analyzed using descriptive statistics. Data normality was tested using skewness and kurtosis. Correlations between measurement variables were analyzed using Pearson's correlation coefficient. The hypothetical model's goodness of fit was tested using the following: $\chi^2$ statistics, standard $\chi^2$ (CMIN/DF, Normed $\chi^2$), standardized root mean square residual, goodness-of-fit index, the normed fit index, the Tucker-Lewis index, the comparative fit index, and root mean square error of approximation. A covariance structure analysis was performed using the maximum likelihood method to determine the model's goodness of fit and test the hypotheses. The statistical significance of the direct, indirect, and total effects of the model was analyzed via bootstrapping. All statistical analyses with $p < .05$ were considered statistically significant.

## Ethical considerations

The study was approved by the institutional review board at Konyang University Hospital (approval number: KYUH 2017-07-011) and conducted in accordance with the Declaration of Helsinki. Questionnaires were placed in the nurses' break rooms for nurses to complete voluntarily. Completed questionnaires were then placed in a collection box in the break room. All participants were provided with an information sheet explaining the study purpose and method, management of collected data, protection of personal information, and participants' right to withdraw from the study. Participants who provided written consent were enrolled in the study.

**Table 1.  Participants' general characteristics (N = 330).**

| Variable | Category | n | % | PSMA | |
|---|---|---|---|---|---|
| | | | | Mean ± SD | t or F (p) Scheffe |
| Gender | Male | 25 | 7.6 | 4.45±0.38 | 1.38 (.168) |
| | Female | 305 | 92.4 | 4.28±0.51 | |
| Age (years) | < 30 | 171 | 51.8 | 4.26±0.50 | 2.54 (.081) |
| | 30–39 | 91 | 27.6 | 4.37±0.51 | |
| | ≥ 40 | 48 | 20.6 | 4.55±0.32 | |
| Length of career (years) | < 5 [a] | 157 | 47.6 | 4.16±0.51 | 5.98 |
| | 5 –< 10 [b] | 58 | 17.6 | 4.42±0.12 | (.001) |
| | 10 –<15 [c] | 33 | 10.0 | 4.44±0.49 | a < b, c, d |
| | ≥ 15 [d] | 82 | 24.8 | 4.39±0.50 | |
| Position | Staff nurse | 222 | 67.3 | 4.27±0.50 | -1.64 (.101) |
| | Manager | 108 | 32.7 | 4.43±0.50 | |
| Experienced a patient safety accident | Yes | 211 | 63.9 | 4.31±0.48 | 1.33 (.182) |
| | No | 119 | 36.1 | 4.21±0.56 | |

SD, standard deviation; PSMA, patient safety management activities.

## Results

### Participants' general characteristics

The valid response rate was 91.7% (N = 330). Participants' demographic characteristics are presented in Table 1. There was a difference in the mean of patient safety management activities between the group with less than 5 years of career experience and the group with more than 5 years of career experience (F = 5.98, $p$ = .001). No other statistical significance of the small group mean according to general characteristics was confirmed (Table 1).

### Verification of normality and validity of the measurement variables

The absolute value of skewness was between 0.40–1.35 and the absolute value of kurtosis was between 0.17–2.03. As the absolute values of skewness and kurtosis did not exceed 3 or 10, respectively, the data satisfied univariate normality [19]. The correlation coefficients for the measurement variables did not exceed .80; therefore, multicollinearity was not a concern. Discriminant validity was established as the average variance extracted for each observed variable was greater than its coefficient of determination ($r^2$) (Table 2) [19].

**Table 2.  Correlations among observed variables.**

| Variable | JF | SMS | AT | SN | PBC | BI | AVE | CR |
|---|---|---|---|---|---|---|---|---|
| | r (p) | r (p) | r (p) | r (p) | r (p) | r (p) | | |
| SMS | -.054 (.427) | | | | | | 0.50 | 0.75 |
| AT | .024 (.852) | .300 (.026) | | | | | 0.62 | 0.83 |
| SN | .040 (.502) | .533 (.009) | .428 (.021) | | | | 0.68 | 0.92 |
| PBC | -.180 (.009) | .234 (.030) | .184 (.011) | .218 (.019) | | | 0.71 | 0.91 |
| BI | -.027 (.569) | .386 (.025) | .319 (.010) | .451 (.008) | .632 (.012) | | 0.69 | 0.90 |
| PSMA | .152 (.021) | .369 (.032) | .337 (.012) | .337 (.028) | .400 (.008) | .617 (.019) | 0.61 | 0.90 |

JF, job factors; SMS, safety management system; AT, attitude; SN, subjective norm; PBC, perceived behavioral control; BI, behavioral intention; PSMA, patient safety management activities; AVE, average variance extracted; CR, construct reliability.

**Table 3. Results of the conceptual model analysis.**

| Endo-genous variable | Exo-genous variable | B | SE | t | P | SMC | Direct B (p) | Indirect B (p) | Total B (p) |
|---|---|---|---|---|---|---|---|---|---|
| **AT** | | | | | | .145 | | | |
| | JF | .050 | .047 | 0.856 | .392 | | .050 (.454) | | .050 (.454) |
| | SMS | .381 | .080 | 5.199 | < .001 | | .381 (.010) | | .381 (.010) |
| **SN** | | | | | | .345 | | | |
| | JF | .077 | .057 | 1.500 | .134 | | .077 (.189) | | .077 (.189) |
| | SMS | .587 | .110 | 8.132 | < .001 | | .587 (.010) | | .587 (.010) |
| **PBC** | | | | | | .101 | | | |
| | JF | -.157 | .080 | 2.789 | .005 | | -.157 (.025) | | -.157 (.025) |
| | SMS | .267 | .129 | 4.022 | < .001 | | .267 (.010) | | .267 (.010) |
| **BI** | | | | | | .500 | | | |
| | AT | .121 | .062 | 2.399 | .016 | | .121 (.029) | | .121 (.029) |
| | SN | .303 | .044 | 6.017 | < .001 | | .303 (.010) | | .303 (.010) |
| | PBC | .560 | .039 | 9.803 | < .001 | | .560 (.010) | | .560 (.010) |
| **PSMA** | | | | | | .381 | | | |
| | PBC | .004 | .030 | 0.062 | .951 | | .004 (.929) | .344 (.010) | .348 (.010) |
| | BI | .614 | .053 | 7.556 | < .001 | | .614 (.010) | | .614 (.010) |

Goodness-of-fit statistics: $\chi^2$ = 639.809 (DF = 288, $p < .001$), $\chi^2$/DF = 2.222, SRMR = 0.077, GFI = 0.872, NFI = 0.886, TLI = 0.925, CFI = 0.933, RMSEA = 0.061.
SE, standard error; SMC, squared multiple correlation; JF, job factors; SMS, safety management system; AT, attitude; SN, subjective norm; PBC, perceived behavioral control; BI, behavioral intention; PSMA, patient safety management activities; DF, degrees of freedom; SRMR, standardized root mean squared residual; GFI, goodness-of-fit index; NFI, normed fit index; TLI, Tucker-Lewis index; CFI, comparative fit index; RMSEA, root mean squared error of approximation.

### Confirmatory factor analysis of the conceptual model

Although the goodness-of-fit and normed fit indices were slightly lower than the required values, we determined that the model showed a good fit considering the other indices (Table 3).

Eight out of the eleven paths were significant (Fig 2). The safety management system showed a significant path to attitude, with an explanatory power of 14.5%. Safety management system showed a significant path to subjective norms, with an explanatory power of 34.5%. Job factors and safety management system showed significant paths to perceived behavioral control, with an explanatory power of 10.1%. Attitude, subjective norms, and perceived behavioral control showed significant paths to behavioral intention, with an explanatory power of 50.0%. Behavioral intention showed a significant path to patient safety management activities, with an explanatory power of 38.1% (Table 3).

Table 3 shows the direct and indirect relevance of the hypothesis model. The safety management system has a direct influence on attitudes and subjective norms. Job factors and the safety management system had a direct influence on perceived behavioral control. Attitude, subjective norms, and perceived behavioral control were directly related to behavioral intention. Perceived behavioral control showed indirect relevance to patient safety management activities, and behavioral intention showed direct relevance to patient safety management activities.

### Discussion

The modified TPB model explained patient safety management activities in the OR relatively well. The explanatory power of the model for behavior was high; adding organizational factors as antecedents to personal factors further increased the model's explanatory power.

The higher the job demands, the lower the perceived behavioral control of patient safety management activities. The physical and cognitive burdens of excessive job demands

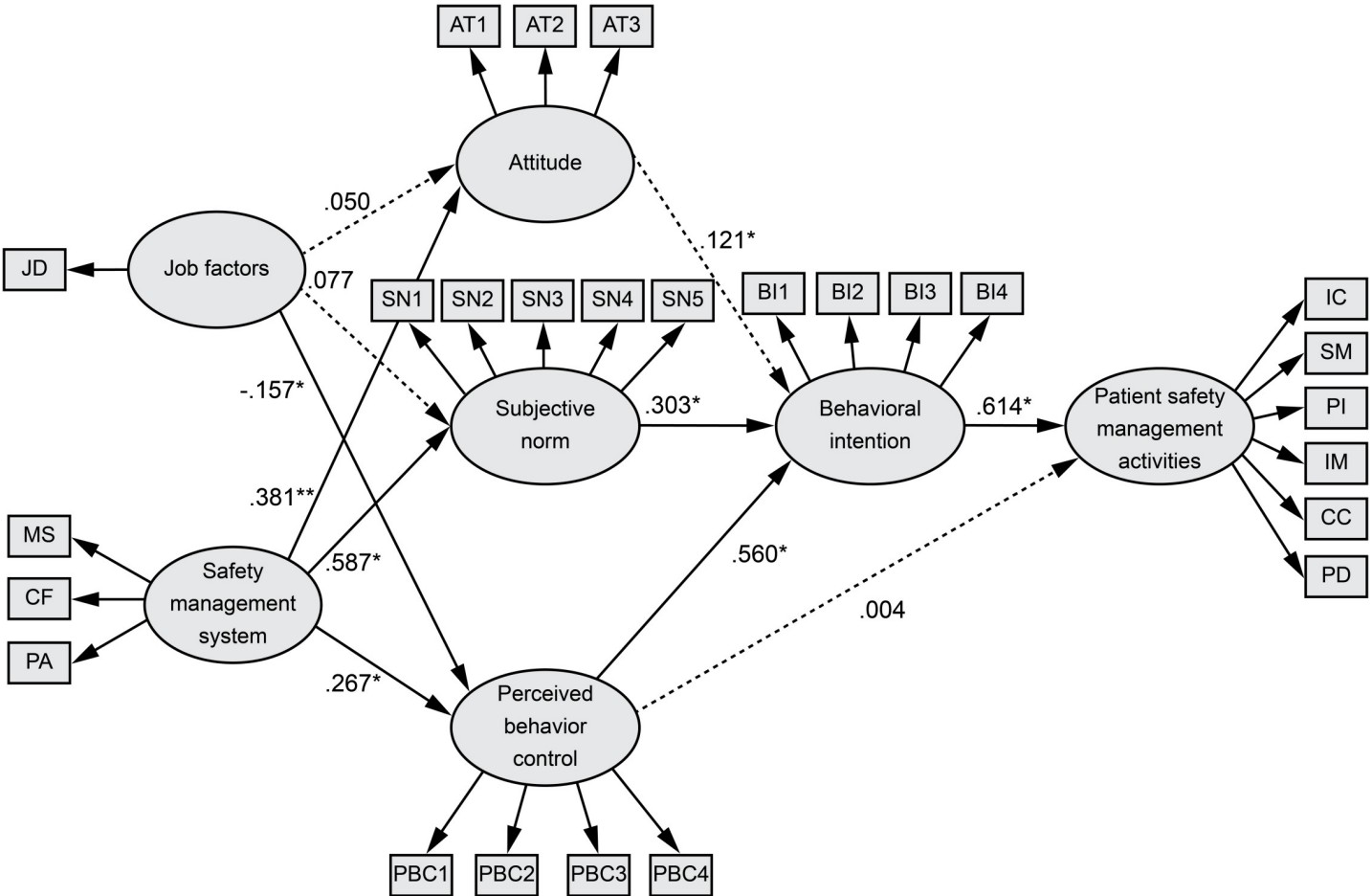

**Fig 2. Path diagram of the model.** JD, job demand; MS, management supervision; CF, communication and feedback; PA, participation; AT, attitude; SN, subjective norm; PBC, perceived behavioral control; BI, behavioral intention; IC, infection control; SM, specimen management; PI, patient identification; IM, item management; CC, count confirmation; PD, prevent damage. *p < .05, **p < .01.

undermine one's problem-solving abilities related to safety performance and are, therefore, associated with increased accident occurrence [27]. Regarding organizational factors, higher scores for safety management system were associated with more positive attitudes, stronger subjective norms, and perceived behavioral control in patient safety management activities. Organizational factors, including management values, the safety system, safety practice, education and training, and communication, could impact individual factors such as safety motivation and knowledge [16]. Further, stronger behavioral intention regarding patient safety management activities was associated with more positive attitudes toward patient safety management activities, stronger subjective norms, and greater perceived behavioral control. These findings were similar to those of previous studies based on the TPB, in which attitude, subjective norms, and perceived behavioral control predicted behavioral intention [28,29]. These results confirmed that the modified TPB is a valid model for explaining patient safety management activities in the OR.

Perceived behavioral control was the most influential factor on the behavioral intention for patient safety management activities in the operating room, followed by subjective norms and attitudes. These results were contrary to a study examining alcohol abstinence in patients with chronic liver disease [30] and one that examined hidden agendas in the use of mental health

services for depression [31]. In these two studies, attitude appeared to have the greatest influence on behavior. In predicting behavioral intention, the influence of attitude, subjective norms, and perceived behavioral control could vary depending on the extent to which behavior and situations are controlled by an individual [32]. In previous studies, factors influencing behavioral intentions showed varying results depending on the characteristics and type of behavior [33]. Unlike individual behavior, perceived behavioral control and subjective norms could be key factors in behavioral intention related to social behavior; these factors are difficult to control through an individual's will alone.

Moreover, the results showed that perceived behavioral control did not directly relate to patient safety management. Thus, perceived behavioral control may not be directly related to behavior if one's perception is not consistent with actual behavioral control; hence, the relationship between behavioral control and behavior is indicated by dotted lines in the TPB [32]. A meta-analysis of the TPB also showed conflicting results for perceived behavioral control depending on the type of behavior involved [28]. Nurses expect to be able to control patient safety management activities, but they may not be able to do so if there are uncontrollable environmental factors such as a heavy workload and a lack of necessary supplies. Conversely, since they may not deliberately perform safety management activities as a result of excessive trust in their skills or reckless behavior, an analysis of the specific path between attitudes, subjective norms, and perceived behavior control needs to be researched. Organizational actions required to improve patient safety management activities in the OR include reducing job demands and enhancing the organizational safety management system. Individual actions required include fostering a positive attitude and increasing one's behavioral intention by strengthening subjective norms and perceived behavioral control. Hospitals should recognize that individuals comprise the organization and devise strategies accordingly to improve patient safety management activities. Specific and practical education tailored to the conditions of the OR should be provided, and standardization of the OR patient safety management protocol and information management are necessary to enhance the efficiency of communication systems.

## Limitations

The data for this study were collected from perioperative nurses working in large hospitals; therefore, future studies should include nurses from small and medium-sized hospitals with varying OR sizes and types of work. In addition, since the research was conducted using self-reported subjective data, casual effects between variables could not be confirmed. Therefore, further research using objective data on factors such as reporting of patient safety accidents (near miss, adverse events, sentinel events) is necessary. Moreover, we established a model based on a modified TPB to explain patient safety management activities in the OR; additional studies that examine other factors associated with patient safety management activities in the OR are needed.

## Conclusions

Crucial influencing factors on patient safety management activities in the OR were the safety management system, subjective norms, perceived behavior control, and behavior intention. Therefore, it is necessary to prepare hospital-level support and nursing policies to reinforce these factors. Organizations as well as individuals and medical staff should work together to strengthen OR patient safety management activities.

## Acknowledgments

We thank the operating room nurses who participated in this study.

## Author Contributions

**Conceptualization:** Sun Young Jeong.

**Data curation:** Sun Young Jeong.

**Formal analysis:** Nam Yi Kim.

**Investigation:** Nam Yi Kim.

**Methodology:** Sun Young Jeong.

**Project administration:** Sun Young Jeong.

**Resources:** Nam Yi Kim.

**Software:** Nam Yi Kim.

**Supervision:** Sun Young Jeong.

**Validation:** Nam Yi Kim.

**Visualization:** Nam Yi Kim.

**Writing – original draft:** Nam Yi Kim.

**Writing – review & editing:** Sun Young Jeong.

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
