## [Decision Letter · Decision Letter 0]

15 Feb 2021

PONE-D-20-31121

Perioperative patient safety management activities: A modified theory of planned behavior

PLOS ONE

Dear Dr. Jeong,

Thank you for submitting your manuscript to PLOS ONE. After careful consideration, we feel that it has merit but does not fully meet PLOS ONE’s publication criteria as it currently stands. Therefore, we invite you to submit a revised version of the manuscript that addresses the points raised during the review process.

The paper addresses an interesting topic. In the revised version of the paper please consider the reviewers' comments listed in the following. Additionally, please consider adding a similar analysis in which you are considering smaller groups created based on the general characteristics in Table 1 and discuss whether there are differences when certain groups are analyzed compared to the whole sample.

We look forward to receiving your revised manuscript.

Kind regards,

Camelia Delcea

Academic Editor

PLOS ONE

Journal Requirements:

2. Please provide the names of all the participating hospitals, and a further details regarding how these hospitals were selected.

Furthermore, Please include additional information regarding the survey or questionnaire used in the study and ensure that you have provided sufficient details that others could replicate the analyses. For instance, if you developed a questionnaire as part of this study and it is not under a copyright more restrictive than CC-BY, please include a copy, in both the original language and English, as Supporting Information.

3. In the submission please specify whether IRB approval was obtained from all participating hospitals.

4.We note that you have indicated that data from this study are available upon request. PLOS only allows data to be available upon request if there are legal or ethical restrictions on sharing data publicly. For information on unacceptable data access restrictions, please see http://journals.plos.org/plosone/s/data-availability#loc-unacceptable-data-access-restrictions.

Reviewers' comments:

Reviewer's Responses to Questions

**Comments to the Author**

1. Is the manuscript technically sound, and do the data support the conclusions?

Reviewer #1: Yes

Reviewer #2: No

2. Has the statistical analysis been performed appropriately and rigorously? 

Reviewer #1: Yes

Reviewer #2: Yes

3. Have the authors made all data underlying the findings in their manuscript fully available?

Reviewer #1: Yes

Reviewer #2: Yes

4. Is the manuscript presented in an intelligible fashion and written in standard English?

Reviewer #1: Yes

Reviewer #2: Yes

5. Review Comments to the Author

Reviewer #1: The research question is interesting and cogent, and the research is based on a sound literature ground concerning safety behaviors. I think the research is worth publishing, but I address some aspects that could be clarified by the authors.

1. Line 83: the authors make reference to the safety management system. It si not clearly explained how these aspects could hinder safety. The authors do not explicitly refer to organizational culture factors, like the kind of leadership, the blame culture, the safety climate (which are implicit in the safety management tool used in the research).

2. Line 125: The authors should explain why they chose the Job Content Questionnaire developed by Karasek et al. \\n

and no other tools. What is the rationale of this choice?

3. Line 138: The 12-item scale is based on an unpublished study by Moon. The authors should provide evidence of the validity of the scale

4. Line 155. Also, the safety management activity instrument, developed by Jeong, is an unpublished research and the authors should provide evidence of its validity

5. Line 279: The issue of behavioral control is controversial. The internal locus of control is generally considered to be a better predictor of safe performance, but in extreme situations it could also represent an excessive trust in one’s own skills and a deliberate exposure to reckless actions. The authors could provide a deeper explanation of these results.

Line 302: the authors mention among the limitations of the research the fact that it was based on self-report data. Self-report tools may be biased by social desirability, especially when they are related to errors, violations, and safety issues. The authors mention objective measurements such as observational surveys, however, I think it could have been useful to add other kind of objective data to the analysis, for instance concerning the rate of adverse events, injuries, near misses, etc.

Reviewer #2: The current article attempts to tackle the important topic of OR safety from the view of OR RNs. The authors provided OR RNs with surveys and matched the results to the TPB model. However the authors' concluded cause and effect from the survey data, rather than acknowledging that survey correlations cannot imply causation and TPB model fit. The authors need re-structure their conclusions to reflect this, rather than assume the survey results fit the TPB model. Essentially, their conclusions outreach the survey data results.

6. PLOS authors have the option to publish the peer review history of their article (what does this mean?). If published, this will include your full peer review and any attached files.

Reviewer #1: **Yes: **Fabrizio Bracco

Reviewer #2: No

---

## [Author Response · Author response to Decision Letter 0]

9 May 2021

Response to Reviewer’s Comments

Reviewer(s)' Comments to Author: 

Reviewer #1: The research question is interesting and cogent, and the research is based on a sound literature ground concerning safety behaviors. I think the research is worth publishing, but I address some aspects that could be clarified by the authors.

1. Line 83: the authors make reference to the safety management system. It si not clearly explained how these aspects could hinder safety. The authors do not explicitly refer to organizational culture factors, like the kind of leadership, the blame culture, the safety climate (which are implicit in the safety management tool used in the research).

Reply: Thank you for your comments. Corrected the sentence. It has been described how the safety management system relates to safety management activities. 

Line 90: The safety management system includes training and participation in safety policy, management supervision, communication, and feedback. Therefore, when the level of safety management system is insufficient, accurate information on safety is not delivered, and education and management are neglected, thus resulting in lower awareness and performance of patient safety management activities [15-17].

2. Line 125: The authors should explain why they chose the Job Content Questionnaire developed by Karasek et al. 

and not other tools. What is the rationale of this choice?

Reply: Thank you for your comments. Corrected the sentence. It was used in a study on the job factors of Korean nurses in previous studies, and the validity and reliability were verified and used in this study. In Song's study, the fitness index of the confirmatory factor analysis for job factors was χ2=88.949 (df=5, p<.001), TLI=.84, CFI=.92, RMSEA=.21, and SRMR=.05.

Line 131: The Job Content Questionnaire developed by Karasek et al. [20] is a commonly used instrument to assess organizational job factors, and its validity and reliability were verified in a previous study by Song [21], which examined job factors for Korean nurses. Therefore, in this study, the instrument modified by Song [21] for perioperative nurses was used.

3. Line 138: The 12-item scale is based on an unpublished study by Moon. The authors should provide evidence of the validity of the scale.

Reply: Thank you for your comments. Corrected the sentence. In the paper published by Moon and Song, the validity of the tool was verified, so the references were revised. 

Line 146: A 12-item scale was developed based on Ajzen's [23] study and a scale developed by Moon and Song [24] for hospital nurses.

4. Line 155. Also the safety management activity instrument, developed by Jeong, is an unpublished research and the authors should provide evidence of its validity

Reply: Thank you for your comments. Corrected the sentence. The reference was changed to a published article.

Line 164: Safety management activities were measured using an instrument developed by Kim and Jeong [25] based on six international patient safety goals [26].

5. Line 279: The issue of behavioral control is controversial. The internal locus of control is generally considered to be a better predictor of safe performance, but in extreme situations it could also represent an excessive trust in one’s own skills and a deliberate exposure to reckless actions. The authors could provide a deeper explanation of these results.

Reply: Thank you for your comments. Additional discussion was written.

Line 294: Nurses expect to be able to control patient safety management activities, but they may not be able to do so if there are uncontrollable environmental factors such as a heavy workload and a lack of necessary supplies. Conversely, since they may not deliberately perform safety management activities as a result of excessive trust in their skills or reckless behavior, an analysis of the specific path between attitudes, subjective norms, and perceived behavior control needs to be researched.

6. Line 302: the authors mention among the limitations of the research the fact that it was based on self-report data. Self-report tools may be biased by social desirability, especially when they are related to errors, violations, and safety issues. The authors mention objective measurements such as observational surveys, however, I think it could have be useful to add other kind of objective data to the analysis, for instance concerning the rate of adverse events, injuries, near misses, etc. 

Reply: Thank you for your comments. Corrected the sentence.

Line 313: In addition, since the research was conducted using self-reported subjective data, casual effects between variables could not be confirmed. Therefore, further research using objective data on factors such as reporting of patient safety accidents (near miss, adverse events, sentinel events) is necessary.

7. The paper addresses an interesting topic. In the revised version of the paper please consider the reviewers' comments listed in the following. Additionally, please consider adding a similar analysis in which you are considering smaller groups created based on the general characteristics in Table 1 and discuss whether there are differences when certain groups are analyzed compared to the whole sample.

 Reply: Thank you for your comments. Further analysis and presented in the results and tables 1.

Line 198: There was a difference in the mean of patient safety management activities between the group with less than 5 years of career experience and the group with more than 5 years of career experience (F=5.98, p=.001). No other statistical significance of the small group mean according to general characteristics was confirmed (Table 1).

Reviewer #2: 

1. The current article attempts to tackle the important topic of OR safety from the view of OR RNs. The authors provided OR RNs with surveys and matched the results to the TPB model. However the authors' concluded cause and effect from the survey data, rather than acknowledging that survey correlations cannot imply causation and TPB model fit. The authors need re-structure their conclusions to reflect this, rather than assume the survey results fit the TPB model. Essentially, their conclusions outreach the survey data results.

Reply: Thank you for your comments. The comment part was added as a limitation of the study, and the discussion and conclusion parts were entirely revised and described.

Line 276: Perceived behavioral control was the most influential factor on the behavioral intention for patient safety management activities in the operating room, followed by subjective norms and attitudes. These results were contrary to a study examining alcohol abstinence in patients with chronic liver disease [30] and one that examined hidden agendas in the use of mental health services for depression [31]. In these two studies, attitude appeared to have the greatest influence on behavior. In predicting behavioral intention, the influence of attitude, subjective norms, and perceived behavioral control could vary depending on the extent to which behavior and situations are controlled by an individual [32]. In previous studies, factors influencing behavioral intentions showed varying results depending on the characteristics and type of behavior [33]. Unlike individual behavior, perceived behavioral control and subjective norms could be key factors in behavioral intention related to social behavior; these factors are difficult to control through an individual’s will alone.

Line 289: Moreover, the results showed that perceived behavioral control did not directly relate to patient safety management. Thus, perceived behavioral control may not be directly related to behavior if one's perception is not consistent with actual behavioral control; hence, the relationship between behavioral control and behavior is indicated by dotted lines in the TPB [32]. A meta-analysis of the TPB also showed conflicting results for perceived behavioral control depending on the type of behavior involved [28]. Nurses expect to be able to control patient safety management activities, but they may not be able to do so if there are uncontrollable environmental factors such as a heavy workload and a lack of necessary supplies. Conversely, since they may not deliberately perform safety management activities as a result of excessive trust in their skills or reckless behavior, an analysis of the specific path between attitudes, subjective norms, and perceived behavior control needs to be researched. 

Line 313: In addition, since the research was conducted using self-reported subjective data, casual effects between variables could not be confirmed. Therefore, further research using objective data on factors such as reporting of patient safety accidents (near miss, adverse events, sentinel events) is necessary. 

Line 322: Crucial influencing factors on patient safety management activities in the OR were the safety management system, subjective norms, perceived behavior control, and behavior intention. Therefore, it is necessary to prepare hospital-level support and nursing policies to reinforce these factors. Organizations as well as individuals and medical staff should work together to strengthen OR patient safety management activities.

---

## [Decision Letter · Decision Letter 1]

20 May 2021

Perioperative patient safety management activities: A modified theory of planned behavior

PONE-D-20-31121R1

Dear Dr. Jeong,

We’re pleased to inform you that your manuscript has been judged scientifically suitable for publication and will be formally accepted for publication once it meets all outstanding technical requirements.

Kind regards,

Camelia Delcea

Academic Editor

PLOS ONE

Additional Editor Comments (optional):

Reviewers' comments:

Reviewer's Responses to Questions

**Comments to the Author**

1. If the authors have adequately addressed your comments raised in a previous round of review and you feel that this manuscript is now acceptable for publication, you may indicate that here to bypass the “Comments to the Author” section, enter your conflict of interest statement in the “Confidential to Editor” section, and submit your "Accept" recommendation.

Reviewer #1: All comments have been addressed

2. Is the manuscript technically sound, and do the data support the conclusions?

Reviewer #1: (No Response)

3. Has the statistical analysis been performed appropriately and rigorously? 

Reviewer #1: (No Response)

4. Have the authors made all data underlying the findings in their manuscript fully available?

Reviewer #1: (No Response)

5. Is the manuscript presented in an intelligible fashion and written in standard English?

Reviewer #1: (No Response)

6. Review Comments to the Author

Reviewer #1: (No Response)

7. PLOS authors have the option to publish the peer review history of their article (what does this mean?). If published, this will include your full peer review and any attached files.

Reviewer #1: **Yes: **Fabrizio Bracco

---

## [Editor Report · Acceptance letter]

18 Jun 2021

PONE-D-20-31121R1 

Perioperative patient safety management activities: A modified theory of planned behavior 

Dear Dr. Jeong:

I'm pleased to inform you that your manuscript has been deemed suitable for publication in PLOS ONE. Congratulations! Your manuscript is now with our production department. 

Kind regards, 

on behalf of

Dr. Camelia Delcea 

Academic Editor

PLOS ONE